# Associations Between Polygenic Risk for Alzheimer’s Disease and Grey Matter Volume Are Dependent on APOE, Pathological and Diagnostic Status

**DOI:** 10.3390/genes16101128

**Published:** 2025-09-25

**Authors:** Valerio Nocella, Riccardo Manca, Annalena Venneri

**Affiliations:** 1Department of Medicine and Surgery, University of Parma, 43125 Parma, Italy; valerio.nocella@studenti.unipr.it (V.N.); riccardo.manca@unipr.it (R.M.); 2Department of Neuroscience, University of Sheffield, Sheffield S10 2HQ, UK

**Keywords:** Alzheimer’s disease, polygenic risk score, APOE ε4 allele, atrophy, β-amyloid

## Abstract

Background/Objectives: Studies have shown that higher polygenic risk scores (PRSs) for Alzheimer’s disease (AD) are associated with smaller volumes in temporal brain regions typically affected by this disease. These effects have also been found in cognitively unimpaired (CU) older adults. This study aimed to investigate the relationship between PRSs and brain volumes in specific areas associated with early AD. Methods: 342 participants were selected from the Alzheimer’s Disease Neuroimaging Initiative and stratified into three groups: 114 amyloid-positive atrophic (A+N+), 114 amyloid-negative non-atrophic (A−N−), and 114 amyloid-positive non-atrophic (A+N−) people. Linear regressions were performed within each group to investigate associations between PRSs and regional grey matter volumes. Analyses were also repeated after stratifying groups by APOE status and clinical diagnosis. Two sensitivity analyses were run to investigate the impact of APOE and amyloid status and concordance across biomarkers. Multiplicity was controlled for using the Benjamini–Hochberg false discovery rate (FDR) approach. Results: Negative associations were observed between PRSs and volumes of the left amygdala and hippocampus in A+N+, right hippocampus in A+N−, and right posterior cingulate cortex in A−N− participants. Associations were found especially in A−N− participants, both ε4 allele carriers and non-carriers, and mostly confirmed in sensitivity analyses. Associations emerged only in CU and AD participants, but not in people with MCI. None of these findings survived correction for FDR. Conclusions: These findings highlight the potential of PRSs as novel biological indicators for a deeper characterisation of AD-related neural alterations.

## 1. Introduction

Alzheimer’s disease (AD) is the most common neurodegenerative disease, affecting about 5% to 7% of the population over 60 years of age, and represents one of the biggest global health challenges, significantly impacting quality of life of patients and their families [1]. AD is characterised by progressive neurocognitive decline, manifested by cognitive and memory deterioration, impairment in daily activities, and a variety of neuropsychiatric symptoms [2].

Diagnostic criteria for AD have evolved considerably over the years. Recent advancements in biomarker research have enabled the inclusion of neuroimaging, plasma, and cerebrospinal fluid (CSF) biomarkers into the diagnostic workup of AD. A variety of biomarkers for amyloid-beta (Aβ) plaques, tau tangles, and neurodegeneration have been proposed [3].

The most common sporadic form of late-onset AD (LOAD) is influenced by a combination of genetic and environmental factors with many genes contributing to determine AD risk [4]. The **ε**4 allele of the apolipoprotein E (APOE) gene is the strongest genetic risk factor for LOAD [5]. However, recent genome-wide association studies (GWAS) have identified additional risk loci, contributing to a more comprehensive understanding of the multifaceted genetics of AD [6] and, thus, providing new insights into possible biological mechanisms underlying this neurodegenerative process [7].

The individual contribution of most genetic variants strongly associated with AD, with the exception of the APOE **ε**4 allelic variant, to increasing an individual’s risk of developing AD appears to be only marginal (1% to 8%) [8]. Polygenic risk scores (PRSs) are calculated by adding the effects of multiple loci across the genome and are used to assess an individual’s genetic predisposition to AD [9,10]. By integrating the cumulative effect of commonly occurring alleles, PRSs may better capture the variance explained by genetic factors [11]. These scores facilitate the identification of individuals at high risk of developing AD, as PRSs are derived from the most recent GWASs [12]. Therefore, PRSs have the potential to enable targeted preventive measures and personalised interventions.

Studies have shown that higher PRSs in cognitively healthy adults are associated with greater cognitive decline [13] and smaller regional brain volumes (e.g., in hippocampus and precuneus) [14,15], especially hippocampal volume, but also that of other brain regions involved in memory and cognitive functions [16]. This has been found in multiple cohort studies in relation to different combinations of single nucleotide polymorphisms (SNPs), usually including the APOE **ε**4 allele among the pool of genes of interest [17,18,19]. Therefore, it is clear that the statistical association between APOE SNPs and AD risk is stronger than that of any other non-APOE SNPs and AD. Hence, investigating the presence of possible associations between PRSs and atrophy in specific areas associated with AD in APOE **ε**4 carriers, but especially in non-carriers, could provide insights into the effectiveness of these scores in predicting early atrophy in high-risk individuals and into the utility of PRS as a biological indicator and as a tool for early detection of the disease.

For these reasons, this study aimed to investigate the association between PRSs for AD and grey matter (GM) volumes of AD-relevant regions of interest in older adults with and without cognitive impairment stratified by Aβ positivity and neurodegeneration status. The first hypothesis was that PRS values would differ significantly between groups, with higher scores observed in the group exhibiting greater global atrophy. It was further hypothesised that higher PRS values would be negatively associated with GM volumes in brain regions characteristically vulnerable to AD pathology burden (e.g., hippocampus, posterior cingulate cortex, parahippocampal gyrus, middle temporal gyrus) and that these associations would be stronger in groups showing evidence of neurodegeneration and among APOE **ε**4 carriers. The second hypothesis was that PRSs could serve as potential predictors of GM variations in the specific brain areas under investigation across the entire sample. Finally, the last hypothesis was that higher PRS values would be associated with smaller brain volumes, particularly in individuals diagnosed with AD dementia and carriers of the APOE **ε**4 allele.

## 2. Materials and Methods

### 2.1. Participants

Data used in this study were obtained from the Alzheimer’s Disease Neuroimaging Initiative (ADNI) database (adni.loni.usc.edu). The ADNI was launched in 2003 as a public–private partnership, led by Principal Investigator Michael W. Weiner, MD. The primary goal of ADNI has been to test whether serial MRI, positron emission tomography, other biological markers, and clinical and neuropsychological assessment can be combined to measure the progression of mild cognitive impairment (MCI) and early AD. All ADNI participants provided written informed consent, and study protocols were approved by each participating site’s institutional review board. For research governance and compliance with ethical standards and informed consent, please consult the ADNI website at www.adni-info.org and associated material. No additional local ethical approval was required, as the ADNI database contains only anonymized, publicly accessible data.

Participants were included based on the availability of genetic, MRI, cognitive, and Aβ data. The lack of any of the above-mentioned assessments represented exclusion criteria for this study.

From an initial sample of 747 participants with available genetic data to calculate PRSs, individuals were categorised into three groups based on Aβ positivity (either A+ or A−) and evidence of neurodegeneration (either N+ or N−), matching the resulting subgroups as closely as possible for sex, age, and education. Participants were classified as A+ if they had either a cerebrospinal fluid Aβ1-42 level equal to or below 977 pg/mL [20] or an amyloid-PET standardised uptake value ratio equal or above 1.11 [21]. Neurodegeneration status was determined based on GM fraction (GMF; see Section 2.4 “MRI data and pre-processing” section for details): participants with GMF values < 1.5 standard deviations from the mean of the cognitively unimpaired group were classified N+. Three groups of 114 participants each were formed by stratifying the sample for Aβ and neurodegeneration and matching participants for age, education and sex using the matchit function from the MatchIt (version 4.5.0) R package:

The A+N+ group consisted of 33 females and 81 males; 6 participants were cognitively unimpaired, 48 had MCI, and 60 had a diagnosis of AD dementia.

The A+N− group consisted of 30 females and 84 males; 25 participants were cognitively unimpaired, 70 had MCI, and 19 had a diagnosis of AD dementia.

The A−N− group consisted of 32 females and 82 males; 53 participants were cognitively unimpaired, 55 had MCI, and 6 had a diagnosis of AD dementia.

There were no participants with evidence of neurodegeneration who were Aβ negative.

Because there were a total of 114 participants with an A+N+ profile, equally sized samples with A−N− and A+N− profiles were created by selecting individuals from the larger pool of ADNI participants with available genetic data, following a close matching procedure for age, sex, and education (without replacement). Participants in the A−N− (*n* = 123) and the A+N− (*n* = 253) subgroups not selected by the matching procedure were excluded from the analytical sample (see Figure 1). T1-weighted MRI images of a participant typical of each group (based on median GMF values) are included in Appendix A.

### 2.2. APOE Genotype

Apolipoprotein E (APOE) genotype status for all participants was available in the ADNI database. For this study, only the condition of being a carrier or non-carrier of the APOE **ε**4 allele was considered. In the overall dataset, there were 199 participants who were **ε**4 non-carriers, and 143 who were **ε**4 carriers. In the three subgroups, in the atrophic-positive subgroup there were 45 **ε**4 non-carriers and 69 **ε**4 carriers, among the normal-positive there were 54 **ε**4 non-carriers and 60 **ε**4 carriers and in the normal-negative group, 100 were **ε**4 non-carriers and 14 **ε**4 carriers.

### 2.3. PRS Calculation

Genotyping was carried out by ADNI using an Illumina OmniExpress array [22]. Genotype data were curated to extract common high-quality autosomal markers using PLINKv2.0 [23]. Quality control parameters were a 90% call rate, 5% minor allele frequency and Hardy–Weinberg equilibrium mid-*p* value of 10^−6^. A total of 1.3 million SNPs passed quality control. From the quality-controlled genotype data, 5 genetic principal components (PCs) were generated using PC-AiR [24] to be used as covariates in the analyses.

A PRS for AD was calculated using a training set and GWAS summary statistics [25]. Only SNPs with imputation information content scores greater than 0.9 were used and duplicate SNPs were removed. To our knowledge, participants in ADNI were not included in any of the discovery GWAS used to calculate PRSs for AD.

PRSs were generated with a Bayesian approach using continuous shrinkage priors (parameters: a = 1, b = 0.5, phi learnt from the dataset) [26] in line with a previous publication [27]. After merging the data with a linkage disequilibrium reference based on the 1000 Genome EUR samples (reference panel: ldblk_1kg_eur), 455,027 SNPs were retained. From each of those SNPs a Bayesian posterior effect size was calculated. Finally, posterior effect sizes were used to calculate PRS values in PRSice v2 [28] without pruning, using two *p*-value thresholds: 0.0001 (PRS1) and 0.001 (PRS2). PRS values were standardised by centering on the overall sample mean and dividing by one standard deviation for use in the analysis. Scripts used are included in the Appendix A.

### 2.4. MRI Data and Pre-Processing

All MRI data were acquired as specified in the ADNI MRI protocol [29], using either 1.5 T or 3 T scanners. Pooling of MRI data acquired at different magnetic field (MF) strengths has previously been shown to be a valid approach, with minimal effects on regional volume quantification [30,31,32]. The steps of the most up-to-date standard voxel-based morphometry (VBM) protocol [33] were carried out using Matlab (Mathworks Inc., Cambridge, UK) and Statistical Parametric Mapping (SPM) 12 (Wellcome Centre Human Neuroimaging, London, UK): (1) images were reoriented to the bi-commissural axis; (2) reoriented images were segmented to separate 3 tissue classes, i.e., GM, white matter and cerebrospinal fluid; (3) GM maps underwent affine non-linear registration to the standard International Consortium of Brain Mapping (ICBM) template (European Brains) in the MNI space and subsequently modulated; and finally, (4) normalised images were smoothed with an 8 mm full-width at half-maximum Gaussian kernel [34]. Global volume for each tissue map was quantified using SPM12 and, finally, total intracranial volume (TIV) was calculated for each participant by summing the values of all 3 extracted tissue maps. GMF was calculated by dividing each individual GM volume by their relative TIV. Only GM maps were used to answer the research question of this study.

GM volumes were extracted from 16 regions of interest (ROIs), 8 in the left hemisphere and 8 in the right hemisphere, using the Automated Anatomical Labelling (AAL) atlas 2 [35]. Bilateral ROIs were selected based on the commonly reported neural alterations associated with AD: amygdala, hippocampus, parahippocampal gyrus, middle temporal gyrus, superior temporal gyrus, fusiform gyrus, posterior cingulate cortex, and medial prefrontal cortex. The decision to focus on these regions was based on prior evidence indicating that the influence of a polygenic hazard score for AD was strongest in these areas [36], along with extensive evidence showing that the effect of AD pathology is strong in these regions from an early stage.

### 2.5. Clinical and Cognitive Data

Global cognitive status was assessed with the Mini Mental State Examination (MMSE) [37]. This test consists of simple questions assessing seven different cognitive areas: orientation to time, orientation to space, short-term and long-term memory, attention and calculation, language, and constructive abilities. The total score ranges between a minimum of 0 and a maximum of 30 points.

### 2.6. Data Analysis

All the analyses were carried out using R www.r-project.org (accessed on 2 May 2025).

Demographic, clinical, and genetic data were compared across groups. To compare age, and education, between groups, Kruskal–Wallis and Dunn post hoc tests with Bonferroni correction were used. To compare sex and APOE load between groups, **χ**^2^ tests were used. And finally, to compare diagnosis between groups, **χ**^2^, and post hoc z-test with Bonferroni correction were used.

PRSs were compared between the groups using the Kruskal–Wallis test and Dunn’s post hoc test with Bonferroni corrections. Subsequently, within each group, the association between PRSs and regional GM volumes was investigated (controlling for PCs, education, age, sex, TIV and MF) with robust linear regressions. The regressions were repeated following stratification by APOE **ε**4 carrier/non-carrier status.

Subsequently, considering the whole sample, possible predictors of volumetric variations in the specific brain areas under consideration were tested with a robust linear model (rlm function of the MASS package in R) including PRSs, 5 PCs, education, sex, and age as predictors (*p*-values were calculated from t-values using Student’s distribution). Additionally, robust linear regression models were applied to investigate the association between PRSs and regional brain volumes, following participant stratification by clinical diagnosis and APOE carrier status, and controlling for age, education, sex, TIV and MF. To ascertain further the impact of APOE relative to other SNPs, a sensitivity analysis was run by using two additional PRSs calculated by excluding the APOE region using the hg19 coordinates chr19 from 44,400,000 to 46,500,000 [38,39]: PSR1_noAPOE_ (*p* = 0.0001) and PRS2_noAPOE_ (*p* = 0.001).

Finally, a second sensitivity analysis was carried out by re-running all models in a sub-sample of participants with amyloid status concordant across CSF- and PET-derived biomarkers (*n* = 218). This analysis was carried out because, in this study, amyloid status was determined using either CSF or PET data depending on availability, but agreement between these biomarkers is not consistently observed across all individuals undergoing testing.

Multiple comparisons were statistically controlled for by applying the Benjamini–Hochberg false discovery rate (FDR) correction jointly across all ROI-level tests, groups and both PRSs (m = 96).

R scripts used for data analysis are included in the Appendix A.

## 3. Results

The demographic, clinical, and genetic characteristics of the groups are summarised in Table 1.

### 3.1. PRSs Across Groups

PRS values (at both thresholds, i.e., *p* = 0.0001 (PRS1) and 0.001 (PRS2)) were significantly different across groups (PRS1: H = 44.79, df = 2, *p* < 0.001; PRS2: H = 44.06, df = 2, *p* < 0.001). Post hoc tests revealed that the A+N+ group had higher PRSs than the A−N− group (PRS1: *p* < 0.001; PRS2: *p* < 0.001), as did the A+N− group (PRS1: *p* < 0.001; PRS2: *p* < 0.001).

### 3.2. Associations Between PRSs and Regional GM Volume Within Individual Groups

In the A+N+ group, higher PRS1 and PRS2 scores were significantly associated with smaller volumes in the left amygdala and parahippocampal gyrus (Table 2). In the A−N− group, both PRSs were negatively associated with right posterior cingulate cortical volumes, while in the A+N− group both PRSs were negatively associated with the right hippocampal volume.

### 3.3. Associations Between PRSs and Regional GM Volumes Within Groups Stratified by APOE Genotype

In the A+N+ and A+N− non-carrier groups, PRSs were negatively associated with the left amygdala volume (Table 3). In the A−N− group, among non-carriers, significant relationships were found between both PRSs and the volumes of bilateral superior temporal gyri, and the right posterior cingulate cortex. In carriers, both PRSs were negatively associated with the volume of all ROIs.

### 3.4. Associations Between PRSs and Regional GM Volumes in the Whole Sample

In the whole sample, both PRSs showed significant associations with all regional GM volumes (Appendix A). For all brain regions, age emerged as a highly significant predictor. Sex was negatively associated with the volume of several brain areas, with males having larger regional GM volumes than females. While MF had an impact on some brain regions, education was not associated with any of the GM regional volumes.

### 3.5. Associations Between PRSs and Regional GM Volumes Stratified by Diagnosis and APOE Carrier Status

No significant associations between either PRSs and regional brain volumes were detected in the CU and MCI groups. In the AD non-carrier group, significant associations were observed between both PRSs and the volumes of the left amygdala only. No significant associations were detected in AD patients who were APOE ε4 carriers (Table 4).

### 3.6. ROI-Wise Associations Between AD PRS and Regional Grey Matter Volume with Joint FDR Across ROIs, Groups, and PRSs

After applying FDR correction, no association reached significance level. Full q-values are reported in Appendix A.

### 3.7. Sensitivity Analysis—PRSs Without APOE

After excluding the APOE region, the PRS1_noAPOE_ was negatively associated with the left amygdala volume in the A+N+ group, while both PRSs were negatively associated with lateral temporal, prefrontal and posterior cingulate volumes in the A−N− group (Appendix A).

After stratification by APOE genotype, higher PRSs were associated with smaller volumes in lateral temporal and prefrontal cortices, in A−N− non-carriers, while associations were widespread across all ROIs in A−N− carriers (Appendix A). In A+N− carriers, both PRSs were negatively associated with the volume of the fusiform gyrus bilaterally.

In the whole sample, PRSs without APOE were negatively associated with volumes of both medio-temporal and lateral temporal ROIs (Appendix A).

After stratification by diagnosis and APOE genotype, PRSs were negatively associated with superior temporal volumes, in CU non-carriers, and across most ROIs apart from the hippocampi, in CU carriers (Appendix A).

None of these associations survived FDR correction (Appendix A).

### 3.8. Sensitivity Analysis—Aβ Positivity

In total 218 participants had amyloid status (either Aβ+ or Aβ−) concordant across CSF and PET examinations: 88 (77.2%) A−N−, 56 (49.1%) A+N- and 74 (64.9%) A+N+ participants.

Higher PRS values (both with and without APOE) were generally associated with smaller volumes in medio-temporal ROIs in the A+N+ and A+N− groups, and in the posterior cingulate in the A−N− group (Appendix A). Similar results were found when groups were stratified by APOE genotype, with the most consistent negative associations found between PRSs without APOE and temporal ROIs in the A+N+ group and in the A−N− non-carrier group (Appendix A).

In the whole group, negative associations were found between PRSs with APOE and all volumes of all ROIs and between PRS without APOE and temporal ROIs, as in the main analysis (Appendix A).

After stratification by diagnosis and APOE genotype, negative associations were found between PRSs without APOE and volumes in various temporal ROIs in CU and AD groups, both in carriers and non-carriers (Appendix A), but not in people with MCI.

None of these associations survived FDR correction (Appendix A).

## 4. Discussion

The results of this study showed that both the A+N+ and A+N− groups had significantly higher PRSs compared with A−N− participants. This aligns with our initial hypothesis and is consistent with the literature reporting negative associations between PRSs for AD and CSF Aβ levels (e.g., [40,41]) and brain atrophy (e.g., [16,42]). More specifically, these findings suggest that a PRS for AD is predictive of A+ status irrespectively of disease severity.

All PRS-volume findings were also evaluated applying a joint Benjamini–Hochberg FDR correction across ROIs, groups, and PRSs. Under this conservative statistical approach, no association met q < 0.05, so the effects described below are interpretable as directional trends (see Appendix A). This joint FDR correction prioritises specificity at the cost of power, potentially attenuating detectable effects.

Regression analyses carried out within individual groups found negative associations between PRSs inclusive of APOE and the volume of medio-temporal regions, where the strongest effects of AD pathology are typically detected [43,44,45,46], in the left hemisphere for A+N+ participants, and in the right hemisphere for the A+N− group. In the A−N− group, relationships were found between PRSs and the volume of the right posterior cingulate cortex, i.e., the brain region that first shows signs of dysfunction in the earliest stages of AD [47]. Similar results were found for PRSs without APOE, although negative associations were widespread in the left posterior cingulate and lateral temporal and medial prefrontal areas, bilaterally, for the A−N− group. In the Aβ sensitivity analysis, fewer associations were detected in general, yet findings were confirmed in left medio-temporal regions (amygdala and hippocampus) in the A+N+ and A+N− groups, and in right superior temporal and posterior cingulate cortices in the A−N− group.

Unexpected results emerged when replicating these analyses after stratifying the three groups by APOE genotype (ε4 carriers vs. non-carriers). For both the A+N+ group and A+N−, only weak associations between PRSs with APOE and the left amygdala volume were found following stratification. In the A−N− group, negative associations were observed between both PRSs and the volume of all brain areas in carriers, and in bilateral superior temporal and right posterior cingulate regions in non-carriers. Very similar findings emerged for PRSs without APOE: negative associations with bilateral fusiform gyrus volumes in the A+N− group, widespread impact on most brain regions in the A−N− carriers, and negative associations with bilateral superior temporal and medial prefrontal areas in the A−N− non-carriers. In the Aβ sensitivity analysis, associations were also found in primarily left-lateralised temporal regions in the A+N+ and A+N− groups. This may have been the result of a reduction in statistical power, and fewer A+ than A− participants were retained in the sensitivity analysis. However, across all analyses, AD polygenic risk appears to be primarily associated with smaller volumes in bilateral medio-temporal and superior temporal regions, with more consistent effects in the left hemisphere.

Although this contradicted the initial prediction that PRS-GM volume associations would emerge primarily in participants with evidence of neurodegeneration and in ε4 carriers, these results suggest that the influence of AD polygenic risk on brain structure may be more readily detectable in the absence of substantial neuropathological burden. In A−N− individuals brain morphometry is likely shaped predominantly by genetic and developmental factors, allowing for a cleaner expression of polygenic effects. In contrast, in A+N− and A+N+ individuals, disease-related mechanisms (i.e., brain pathological protein accumulation) may be the strongest determinants of structural neural damage, thereby masking or confounding the possible effects of genetic predisposition. Furthermore, it is possible that non-carriers may exhibit brain alterations due to either non-APOE-related genetic or environmental factors (e.g., [48,49,50,51]). In this group, aside from associations in the right posterior cingulate cortex that resembled those observed in carriers, PRSs were negatively associated with the volumes of the bilateral superior temporal gyri, brain regions where AD-related pathological changes typically accumulate at later stages in most patients. However, although Braak and Braak staging [52] indicates that tau spreads into the neocortex after Aβ accumulation, some studies have shown that tau can begin to accumulate widely in neocortical areas even in cognitively healthy, amyloid-negative individuals [53], particularly as part of the ageing process. Subthreshold levels of Aβ can favour tau accumulation [54] and, consequently, brain atrophy, and patterns of cerebral tau pathology also predict the topography and severity of neurodegeneration [55]. Therefore, it is possible that the PRS influence leading to smaller GM volume in the temporal regions of the A−N− non-carrier group might be driven primarily by tau pathology [56].

The negative trends between PRSs and volumes of lateral temporal regions are not surprising. Atrophy in the middle temporal gyrus, for instance, has been highlighted as a predictor of decline from normal cognition to AD dementia [57]. The left superior temporal gyrus, on the other hand, is a region affected by AD and plays a key role in language [58], a cognitive function used as a diagnostic tool in the early stages of the disease [59,60].

Additionally, a negative association between PRSs and the volume of the posterior cingulate cortex appears to be particularly interesting considering the early involvement of this area in the pathophysiological process of AD [43,61,62]. In A−N− APOE ε4 carriers, a significant association between PRSs and the volume of both hippocampi can be explained based on available evidence that hippocampal atrophy has also been observed in the absence of amyloid positivity [63], and the suggestion that it could be due to different pathogenetic processes [64]. Moreover, the observed significance in the parahippocampal gyrus may be explained by the fact that atrophy in this region is considered an early biomarker of AD [65], similarly to what is observed in the fusiform gyrus, where being a carrier also results in an accelerated rate of atrophy [66].

In the whole sample, negative associations were found between the PRSs with APOE and all regional brain volumes. When the APOE region was excluded, associations with volumes of the posterior cingulate and the fusiform gyrus, bilaterally, and of the right medial prefrontal cortex were no longer significant. All these results were largely replicated in the Aβ sensitivity analysis. Overall, this pattern of findings further suggests that AD polygenic risk, even independently of APOE, may influence the dimensions of a wide range of brain regions usually affected by this disease.

When the sample was stratified by clinical diagnosis, PRSs inclusive of APOE were negatively associated with the left amygdala volume in non-carrier individuals with AD dementia only. PRSs without APOE were negatively associated with regional GM volumes in the CU groups: in the bilateral superior temporal gyrus only, in non-carriers; and in bilateral amygdala, posterior cingulate, lateral temporal and medial prefrontal areas, in carriers. Similar associations were detected in the sensitivity analysis, with some weak associations also emerging between PRSs inclusive of APOE and the volumes of medial and lateral temporal structures in CU and AD carriers. Overall, these findings are consistent with the known trajectory of neurodegeneration in AD, where the medio-temporal lobe structures are among the regions that are most strongly affected by pathological changes [52,67]. This pattern suggests that cumulative genetic risk may worsen GM loss even after clinical onset. In preclinical stages, non-APOE genetic risk may contribute to smaller GM volumes in lateral temporal regions bilaterally, whereas APOE appears to exert a stronger influence that also extends to the medial temporal and posterior cingulate regions. Indeed, the genetic risk conferred by the APOE ε4 allele across the lifespan strongly affects rate of atrophy [5], whereas in non-carriers, the cumulative effect of several other risk loci may become more prominent and captured by a PRS. Moreover, the consistency of the results between PRS1 and PRS2 strengthens the robustness of these findings, even though none survived FDR correction, and is suggestive of a biological vulnerability in medio-temporal areas (supporting memory, emotion and cognitive processing [68]), that is detected by PRSs derived using different statistical thresholds. In fact, it has long been established that the brain regions most strongly associated with the AD PRS in the present study (i.e., hippocampus, parahippocampal gyrus, amygdala, and posterior cingulate cortex) are critically involved in the early manifestation of AD, as consistently reported in the literature [44,61,69,70,71].

Among the demographic covariates included in all statistical models, age was negatively associated with volumes of all brain regions. This effect was expected, since, apart from being a risk factor for AD, older age is naturally associated with brain volume loss [72]. Sex was also a strong predictor of brain volume across many regions, especially bilateral medio-temporal and fusiform areas and the right posterior cingulate regions. In detail, women had smaller volumes in all regions. It must be mentioned that PRSs, either with or without APOE, did not differ between men and women. Even though potential sex-related differential effects have not been tested in this study, a substantial proportion of volumetric variability may be unrelated to AD genetic risk and influenced by non-genetic factors. Contrary to expectations, education was not associated with the volume of any of the regions investigated. Since education is usually considered a proxy measure of cognitive reserve, it might be more strongly associated with brain functioning (not tested in this study) rather than with brain structure [73].

This study has some limitations. First, information on amyloid was variable across participants, as some had only either CSF or PET data. Second, APOE genotype was treated as a binary variable, due to the small number of participants with rarer genotypes (e.g., those carriers of the ε2 allele), thus potentially missing more specific APOE-mediated effects. Third, the lack of control over environmental factors (e.g., diet, physical activity, exposure to pollutants, etc.) that are known to influence both AD and brain damage risks could have attenuated estimates of the real strength of the PRS influence. Fourth, although this study observed directional associations between higher PRSs and smaller regional GM volumes (consistent with prior literature), the sample consisted of ADNI participants predominantly of White European ancestry (110/114 A−N−, 106/114 A+N−, 108/114 A+N+), thus limiting the generalisability of the findings. This is a limitation common to many GWASs and, although having a small number of people from a minority ethnic background in our sample might have introduced variability in the data, not related to AD risk, we decided to retain them, to maximise the power of this study and increase sample diversity. However, these results should be confirmed in future investigations in more diverse cohorts. Fifth, the hypothesis-driven selection of brain areas might have overlooked other structural alterations influenced by a PRS for AD. Sixth, the impact of other age- and health-related factors (e.g., cardio-vascular risks, diet, physical exercise) that were not specifically assessed in this study, but that are known to influence brain health, especially via inflammatory pathways, cannot be excluded and should be clarified in future investigations.

## 5. Conclusions

In conclusion, the results of this study support the potential association between a PRS for AD and volumetric alterations in brain areas highly vulnerable to AD neuropathology. Although the findings are heterogeneous across the differently stratified samples by Aβ and neurodegeneration positivity, diagnosis and APOE genotype, the associations observed in critical areas, such as the hippocampus, highlight the importance of PRSs as predictive tools for AD-related pathological changes. Considering the apparent impact of PRSs on key AD areas in A− CU older adults, future longitudinal studies should assess the predictive power of this measure by tracking the rates of progression to AD dementia in these individuals. The observed effects also underline the need to identify more specific and reliable AD biomarkers than those commonly used (e.g., [39]).

Investigations focused on the influence of a PRS for AD in early disease stages would further enhance our understanding of the aetiopathogenesis of this disease. Identifying brain regions more strongly influenced by a PRS for AD in A−N− individuals would shift researchers’ attention towards a deeper understanding of the complex and not completely understood bio-environmental dynamics involved in AD. If validated extensively, PRSs could inform clinical practice for early diagnosis, and pave the way for personalised prevention strategies aimed at intervening before the neurodegenerative process becomes irreversible. Furthermore, expanding research to include other ethnic populations and incorporating more biomarkers could lead to a more accurate integrated predictive model of AD. A multi-biomarker approach would improve the identification of at-risk individuals while monitoring cognitive decline leading to enhanced prognosis and treatment. With advancements in genomic sequencing technologies, expanding public databases, and a better understanding of epigenetic mechanisms, more comprehensive risk models could be built that also account for interactions with environmental factors. Finally, the most important implication might be in prevention. Identifying individuals at risk before symptom onset could prompt them to adopt preventive strategies, such as lifestyle modifications and cognitive interventions, aimed at preventing or delaying the clinical manifestations of the disease.

## Figures and Tables

**Figure 1 genes-16-01128-f001:**
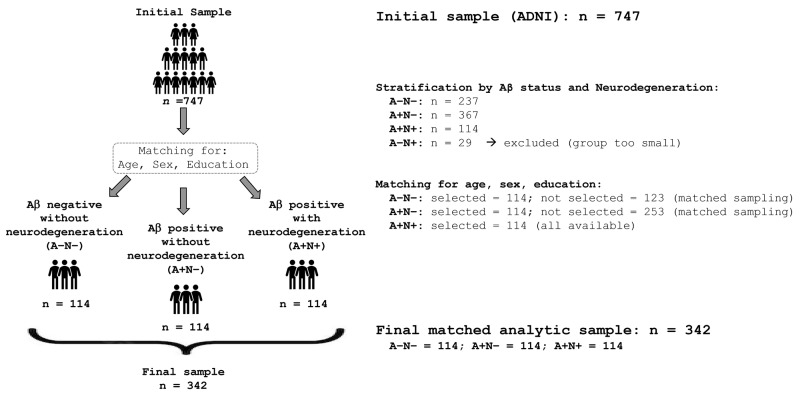
Schematic representation of group selection and matching procedures.

**Table 1 genes-16-01128-t001:** Demographic, clinical, cognitive, and genetic characteristics of the three groups.

Variable	A+N+ (*n* = 114)	A+N− (*n* = 114)	A−N− (*n* = 114)	χ^2^	*p*
Age (years)	78.2 ± 6.8	77.6 ± 6.3	76.5 ± 6.2	4.58	0.101
Education (years)	16.3 ± 2.9	16.2 ± 2.7	16.3 ± 2.7	0.26	0.878
Sex (m/f)	81/33	84/30	82/32	0.20	0.903
Diagnosis (CU/MCI/AD)	64/48/60 ^a,b^	53/55/6 ^a^	25/70/19	100.38	<0.001
APOE (carrier/non-carrier)	69/45 ^a^	60/54 ^a^	14/100	62.76	<0.001

CU: cognitively unimpaired; MCI: mild cognitive impairment; AD: Alzheimer’s disease dementia; APOE: Apolipoprotein E. ^a^ Significantly different from the A−N− group. ^b^ Significantly different from the A+N− group.

**Table 2 genes-16-01128-t002:** Significant associations between PRSs and regional GM volumes within individual groups.

	PRS1			PRS2		
	β	SE	*p*	β	SE	*p*
**A+N+**						
Left amygdala	−0.025	0.010	0.016	−0.025	0.010	0.017
Left parahippocampal gyrus	−0.067	0.031	0.031	−0.068	0.031	0.028
**A−N−**						
Right posterior cingulate cortex	−0.045	0.015	0.003	−0.046	0.014	0.002
**A+N−**						
Right hippocampus	−0.061	0.029	0.039	−0.058	0.029	0.047

SE: Standard error.

**Table 3 genes-16-01128-t003:** Significant associations between PRSs and regional GM volumes within groups stratified by APOE genotype.

	PRS1			PRS2		
	β	SE	*p*	β	SE	*p*
**A+N+ non-carriers**						
Left amygdala	n.s.	n.s.	n.s.	−0.040	0.020	0.048
**A−N− non-carriers**						
Left superior temporal gyrus	−0.274	0.127	0.030	−0.281	0.125	0.025
Right superior temporal gyrus	−0.322	0.153	0.036	−0.325	0.149	0.029
Right posterior cingulate cortex	−0.107	0.048	0.025	n.s.	n.s.	n.s.
**A−N− carriers**						
Left amygdala	0.030	0.000	<0.001	0.032	0.000	<0.001
Right amygdala	−0.002	0.000	<0.001	−0.003	0.000	<0.001
Left hippocampus	0.108	0.000	<0.001	0.121	0.000	<0.001
Right hippocampus	0.177	0.000	<0.001	0.202	0.000	<0.001
Left parahippocampal gyrus	0.136	0.000	<0.001	0.155	0.000	<0.001
Right parahippocampal gyrus	0.249	0.000	<0.001	0.284	0.000	<0.001
Left middle temporal gyrus	0.631	0.005	<0.001	0.714	0.003	<0.001
Right middle temporal gyrus	0.204	0.003	<0.001	0.231	0.002	<0.001
Left superior temporal gyrus	0.556	0.001	<0.001	0.627	0.001	<0.001
Right superior temporal gyrus	0.368	0.001	<0.001	0.417	0.001	<0.001
Left fusiform gyrus	0.364	0.000	<0.001	0.403	0.000	<0.001
Right fusiform gyrus	0.026	0.000	<0.001	0.028	0.000	<0.001
Left medial prefrontal cortex	0.032	0.000	<0.001	0.036	0.000	<0.001
Right medial prefrontal cortex	0.111	0.000	<0.001	0.124	0.000	<0.001
Left posterior cingulate cortex	−0.118	0.000	<0.001	−0.132	0.000	<0.001
Right posterior cingulate cortex	0.020	0.000	<0.001	0.023	0.000	<0.001
**A+N− non-carriers**						
Left amygdala	−0.040	0.017	0.020	n.s.	n.s.	n.s.

n.s.: Not significant, SE: Standard error.

**Table 4 genes-16-01128-t004:** Significant associations between PRSs and brain regional volumes in the AD non-carrier group.

	PRS1			PRS2		
	β	SE	*p*	β	SE	*p*
Left amygdala	−0.048	0.022	0.025	−0.050	0.022	0.025

SE: Standard error.

## Data Availability

All data used in this study are available from ADNI (https://adni.loni.usc.edu/; accessed on 10 May 2022) upon request.

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
