# Peer review of "Associations Between Polygenic Risk for Alzheimer’s Disease and Grey Matter Volume Are Dependent on APOE, Pathological and Diagnostic Status"

_genes, 2025, doi:10.3390/genes16101128_

Round 1

Reviewer 1 Report

Comments and Suggestions for Authors

The authors present an interesting study in which the polygenic risk scores for a number of patient populations was examined in relation to brain volumes in specific areas. Briefly, the authors utilised data from a biobank of patient information and compiled three groups based on their amyloid beta protein status and also level of neuronal atrophy. The values for these data where then cross-referenced against data on brain volume measurements obtained in several regions of the brain. This data was further scrutinised against the presence of the APOE allele e4. In conclusion, the authors ultimately found that there were interesting correlation between PRS and volumes of the hippocampus, posterior cingulate cortex amongst others, and these differences were associated with particular patient profiles. Ultimately, the authors present strong evidence that these PRS scores can act as novel biomarkers for characterising the Alzheimer’s profile of an individual, and warrants further investigation.

In reviewing he manuscript I made a couple of observations. The following should be considered by the authors when preparing a suitable revision.  

  1. For Table 3, why is it that the ‘left middle temporal gyrus’ group has ‘n.s’ for the values for PRS2 when all other groups have values whether significant or not.
  2. In Table 4, there are abbreviations included under the table which are not used/not applicable in this instance and they should be removed/located where appropriate.
  3. As this analyses is done based on measurements taken form images, is it possible to include representative images of each experimental group, or were these values precalculated/this data is not available? It would be beneficial to a reader if even included as an appendix for context.

Reviewer 2 Report

Comments and Suggestions for Authors

The article examines the association between polygenic risk for Alzheimer’s disease (PRS) and gray-matter volume in selected ROIs in ADNI participants, accounting for amyloid status, neurodegeneration, diagnosis, and APOE status. PRS effects are mainly visible in stratified analyses (e.g., A+N+ and some ε4 non-carriers). In the full sample the effects are weaker.

  • Please report PRS-CS hyperparameters and LD settings, and add a sensitivity excluding the APOE region
  • Apply and report FDR (q-values) or another correction across ROIs, strata, and PRS variants.
  • Clarify whether ROI models adjust for TIV. Add sensitivity to scanner field strength (1.5T vs 3T).
  • The paper includes a schematic of selection, please expand Figure 1 to add explicit inclusion/exclusion counts and reasons.
  • Explain how you combined CSF and PET. Add a sensitivity analysis restricted to cases with concordant modalities.
  • Justify the choice of 16 ROIs. Consider bilateral composites or report lateralization explicitly to reduce the number of tests.
  • Share scripts and parameters (R code, PRS pipeline) to enable reproducibility.
  • Motivate the 8-mm FWHM kernel and state the modulation type (non-linear vs affine+non-linear). Provide any additional preprocessing parameters.
  • Provide means±SD.
  • Use consistent anatomical terms (e.g., “medial temporal”), group labels, and abbreviations throughout.
  • Expand the discussion on potential confounders (age and sex dominate effects), sample selection, ancestry homogeneity, and alternative explanations (e.g., vascular or inflammatory components).
  • The in-text citations and the reference list do not follow the journal’s required format. Please align both the in-text citation style and the References section with Genes (MDPI) “Instructions for Authors,” including ordering, punctuation, journal abbreviations, and DOI formatting.

Round 2

Reviewer 2 Report

Comments and Suggestions for Authors

I would like to thank the authors for addressing all the comments. I have no further questions or concerns, and I recommend acceptance of the manuscript in its current form.